# Locomotion in virtual environments predicts cardiovascular responsiveness to subsequent stressful challenges

João Rodrigues [1✉], Erik Studer[1], Stephan Streuber[1], Nathalie Meyer[1] & Carmen Sandi [1✉]

Individuals differ in their physiological responsiveness to stressful challenges, and stress potentiates the development of many diseases. Heart rate variability (HRV), a measure of cardiac vagal break, is emerging as a strong index of physiological stress vulnerability. Thus, it is important to develop tools that identify predictive markers of individual differences in HRV responsiveness without exposing subjects to high stress. Here, using machine learning approaches, we show the strong predictive power of high-dimensional locomotor responses during novelty exploration to predict HRV responsiveness during stress exposure. Locomotor responses are collected in two ecologically valid virtual reality scenarios inspired by the animal literature and stress is elicited and measured in a third threatening virtual scenario. Our model's predictions generalize to other stressful challenges and outperforms other stress prediction instruments, such as anxiety questionnaires. Our study paves the way for the development of behavioral digital phenotyping tools for early detection of stress-vulnerable individuals.

[1] Laboratory of Behavioral Genetics, Brain Mind Institute, School of Life Sciences, École Polytechnique Fédérale de Lausanne, EPFL, Lausanne 1015, Switzerland . ✉email: joao.rodrigues@epfl.ch; carmen.sandi@epfl.ch

Exposure to psychogenic stressors is associated with the development of numerous diseases[1], including psychopathologies[2–5] and cardiovascular disorders (CVD)[6–10]. Through coordinated actions of the autonomic nervous system (ANS) and the hypothalamic–pituitary–adrenal (HPA) axis under the regulation of limbic circuits, acute stress responses support organisms' adaptation to environmental demands[11], but repeated activation of these physiological effectors can be highly detrimental to health[12,13]. However, not all individuals are equally vulnerable to the development of psychopathologies[1,14] or CVDs[6,10,15] following stress exposure. Thus, the identification of stress-susceptible individuals is a critical objective in the establishment of effective disease prevention strategies. Importantly, given that experimental exposure to highly stressful conditions may be detrimental to health in susceptible individuals, it is paramount to develop diagnostic tools that can identify biomarkers predicting variation in stress vulnerability without the need to expose subjects to strong challenges.

The discovery of noninvasive biomarkers for disease is a main goal in contemporary biomedicine[16]. A key emerging question is whether high-dimensional behavioral data can be used for clinical diagnosis[17,18], particularly in the domain of mental health[19–22]. Instead of relying on the use of ratings by clinicians or self-reports from patients, the goal is to use objective and quantifiable behavioral data combined with artificial intelligence analyses to predict disease vulnerability or progression. Passively acquired daily locomotor data (i.e., not involving manual annotation) through room sensing technologies or wearables (i.e., digital phenotyping) have been started to be used to predict disease progression in elderly populations[23–25] or in individuals at risk for self-reported anxiety, depression or stress (for a review see ref. [21]). However, to our knowledge, no study has demonstrated the prognostic value of behavioral data as a predictor of stress vulnerability using objective (i.e., not self-reported) stress measurements.

Given the heterogeneity of life challenges and conditions among human subjects, establishing a diagnostic value of real-life behavioral data from passive digital phenotyping approaches to predict stress vulnerability is still extremely challenging. As an a priori step, substantial progress can be made by probing subjects under experimentally controlled situations, such as immersion in virtual reality (VR) environments. Immersive VR is increasingly used in research and clinics because it is effective in eliciting relevant behavioral, emotional, and physiological responses to different scenarios[26–28], including stressful ones[29–31]. Rodents' locomotor responses in mildly arousing exploration tasks can predict their future vulnerability to stress exposure[32–35]. Inspired by these behavioral phenotyping tasks from the rodent literature, here we aimed to develop immersive virtual environments (IVEs) to predict physiological stress susceptibility.

To define stress vulnerability, we focused on heart rate variability (HRV)[36–38], a surrogate index of cardiac vagal break that yields information about flexibility of the ANS[39]. HRV is a measure of fluctuations over time in cardiac interbeat intervals due to the interaction of the two ANS branches: sympathetic (SNS) and parasympathetic (PNS). In addition, tonic HRV reflects the degree of flexible control exerted by the prefrontal cortex over the periphery[39]. In recent assessments of the literature, high HRV levels during resting conditions have been identified as a top biomarker for stress resilience[40,41]. Conversely, tonic low HRV values—which are indicative of reduced vagal cardiac control—are associated with increased risk of developing psychopathologies[36,42], a broad range of CVDs[8,43] and, more generally, all-cause mortality[44,45] (see Supplementary Notes for further details on the mechanisms underlying HRV and the effectiveness of HRV as a biomarker in stress-related psychopathologies). Stress-related cardiovascular reactivity values elicited in the laboratory setting have been shown to outperform the predictive capacity of resting assessments in studying disease development[6,46–48] and also found to be rather stable personal characteristics that are fairly consistent across time and between stressors[6,49]. However, in the long-term, the prognostic value of stress reactivity studies has proven to be quite modest[50–52], typically based on exposure to short and non-ecologically relevant laboratory stressors, and relying on a handful of variables. Although these limitations have been overcome by exposure to more complex challenges and performing ambulatory cardiovascular measurements[53,54], these approaches are rather expensive, time-consuming and not easy to standardize for interindividual comparisons.

In this work, in order to address these issues, we design three IVEs to which participants are subsequently exposed. The first two IVEs are inspired by classical behavioral phenotyping tasks from the rodent literature (i.e., open field and elevated plus maze), whereas the third IVE engenders persistent threat with the aim to elicit sustained parasympathetic withdrawal (i.e., low HRV). We also compute a robust HRV index devoid of contamination from respiration. By applying machine learning methods, first to a training data set and then to a testing data set, we show that we can predict participants' HRV responses to the third (stressful) scenario from high-dimensional locomotor data obtained during participants' exposure to the first two exploratory IVEs. Importantly, our model is not only validated in a different cohort but it also generalizes across different stressful situations.

## Results

**VR scenarios for behavioral and physiological phenotyping.** Participants, equipped with a head-mounted display and wireless sensors for motion capture and physiology, were asked to explore three consecutive VR scenarios (see Methods). Our goal was to obtain high-density behavioral data from the first two phenotyping scenarios (scenarios 1 and 2) and then feed these data into a gradient tree boosting regression model to predict participants' cardiovascular reactivity when they were subsequently exposed to the persistently threatening scenario (scenario 3). Thus, scenarios 1 (empty room) and 2 (elevated alley) were designed with ecological validity to reveal variance in participants' locomotor responses while they explored mildly arousing conditions, whereas scenario 3 represented a stressful situation with the aim to elicit heightened physiological responses.

The two phenotypic scenarios consisted of an empty room (Fig. 1a) and an elevated alley raised at the level of roofs in a virtual city (Fig. 1b). Although both mimicked key aspects of tests from the rodent literature (see below), our goal was to adapt the scenarios to standard laboratory room dimensions (i.e., 3.5 m × 6 m), which determined that large spaces or redundant aspects of the rodent tests were simplified to their gist. Specifically, the first scenario mimicked the open-field test[55], and the second scenario the open arms aversive elements of the elevated plus maze test[56] and the elevated successive alleys test[57]. High-dimensional behavioral data were obtained from each participant's locomotor responses while exploring scenarios 1 and 2 via the VR headset's position tracking (allowing computation of parameters related to positioning in the virtual space across time, velocity and acceleration for vertical and horizontal movements and trajectory features; Fig. 1c, d) and a lower body motion capture suit (allowing computation of gait variables; Fig 1d, e). In addition, data from both channels allowed us to compute movement burst features and immobility features. A detailed list of all behavioral features used to train our model is included in the Supplementary Information, section "Behavioral feature description", Supplementary Tables 1–5, comprising a complete description of all behavioral parameters. The third

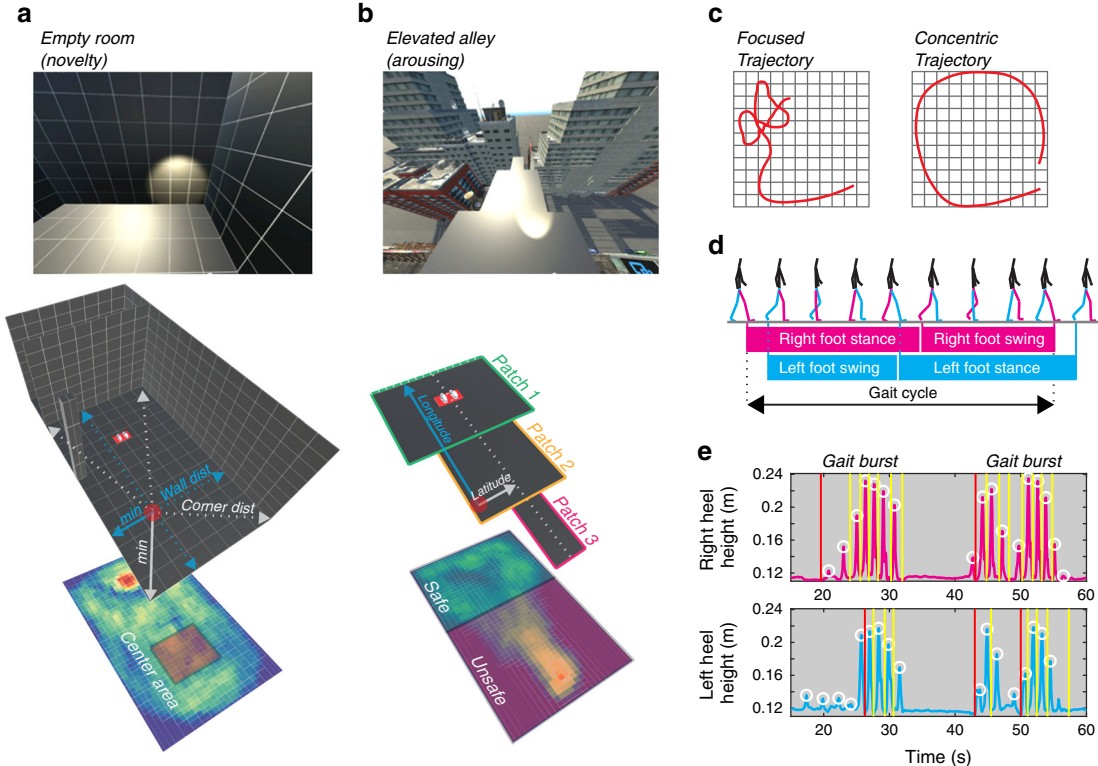

**Fig. 1 Virtual reality scenarios designed to reveal behavioral variance in participants' locomotor responses under nonthreatening conditions.**
**a** Scenario 1: Empty room. Illustrations and example of key features feeding location parameters (e.g., distance to walls, corners, and center area) plotted on an average exploration heatmap. **b** Scenario 2: Elevated alley, consisting of successive narrower patches. Illustration and example of key features feeding location parameters, such as the threshold between patches to determine time to cross or time spent at each patch. **c** Examples of focused and concentric trajectories displayed in scenario 1 (adapted from ref. [101]). **d** Example of the gait cycle and its subcomponents stance and swing. **e** Example of the detection of bursts of movement. Consecutive gait cycles, starting at the red line, separated by immobility periods.

scenario, a dark maze corridor in which startling stimuli were occasionally presented (Fig. 2a), was designed to elicit feelings of persistent threat and, potentially, heightened cardiovascular responses. As shown in Fig. 2b (see also Supplementary Tables 6, 7 for statistics), participants' HRV indeed decreased and their heart rate (HR) increased over time during dark maze exploration, representing marked parasympathetic withdrawal. This lack of cardiovascular adaptation during immersion confirmed the efficiency of this virtual scenario in triggering escalating cardiovascular reactivity. These data support the relevance of this scenario for studying physiological reactions under stress. The reactivity elicited in this scenario clearly differed from that elicited by the phenotyping scenarios, in which participants' HRV root mean square of successive differences (RMSSD) and HR responses progressively habituated during immersion in each VR scenario (Supplementary Fig. 1 and Supplementary Tables 8–11).

**Creation of an integrated HRV index.** From the range of cardiorespiratory markers available [e.g., HR, respiration rate (RR), HRV], we selected HRV as the predictive biomarker to focus on in our study given the importance of HRV to predict vulnerability to disease (see Introduction and Supplementary Information "Cardiorespiratory variables" for exhaustive details on HRV measurements). Several formulas have been developed to calculate HRV[58], and each formula involves a different degree of breathing influence[59]. Therefore, instead of choosing a single formula for model training, we aimed to produce a generalizable and more robust measure of HRV-related parasympathetic prevalence with the most widely used time-domain formulas for parasympathetic activity[60]: the RMSSD, the standard deviation of

the normal-to-normal intervals (SDNN) and the HRV triangular index (HRVTi). To this end, we computed an integrated HRV index (iHRV) using these three HRV formulas together with HR and RR (see Methods and Supplementary Information) obtained while subjects explored the VR scenarios. Specifically, by applying principal component analysis (PCA), we identified a principal component (PC1; hereafter iHRV) that explained most of the variance in our data. Importantly, iHRV loaded positively on all HRV formulas used but negatively on HR, indicating an excellent representation of HRV-related parasympathetic prevalence (Supplementary Fig. 2). In addition, RR loaded on a principal component other than PC1, indicating that PC1 was devoid of contamination from potential differences in participants' respiration. Therefore, using PC1 as the iHRV for model training ensured that our model would not be affected by differences in breathing patterns across subjects or time.

**Using machine learning to predict HRV to stress from behavior.** To develop a gradient tree boosting regression model to predict iHRV from behavioral phenotyping data, we first applied feature selection to 172 behavioral features from scenarios 1 and 2 (i.e., the empty room and the elevated alley, respectively). As a result, we obtained 18 features (see Table S12). To avoid overfitting and to estimate the association between our model's predictions and iHRV, we used a train/test analysis design. Data were split into training/discovery ($N = 66$ subjects, age: $21.00 \pm 1.93$ years) and testing/replication ($N = 69$ subjects, age: $20.19 \pm 2.10$ years) data sets. Given that trait anxiety (as measured with the Trait questionnaire from the State-Trait Anxiety Inventory (STAI-T)) can be associated with HRV[61], the training/discovery

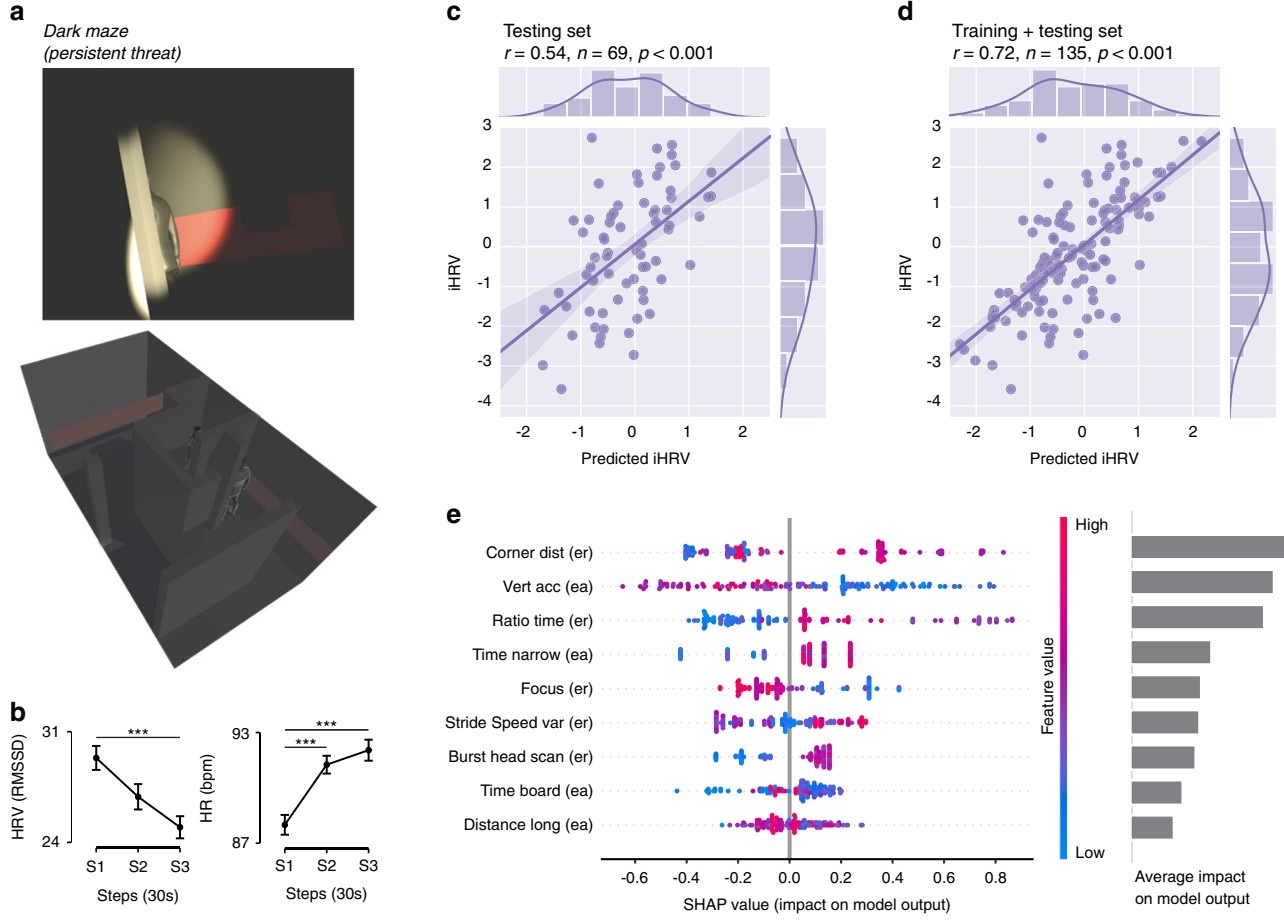

**Fig. 2 Stressful virtual reality scenario and model prediction of cardiac stress responses from behavior. a** Scenario 3: Dark maze, aimed at eliciting heightened physiological responses. **b** Heart rate (HR) and heart rate variability (HRV, the RMSSD formula) over time (three blocks of 30 s each) during dark maze exploration. $N = 135$ participants examined over three consecutive blocks of 30 s (S1, S2, and S3). Data are presented as mean values ± SEM (vertical lines). A repeated measures analysis of variance (rm-ANOVA) was performed for both HR and HRV data sets. Post hoc tests were performed with two-sided paired $t$ tests, with $p$ values corrected for multiple comparisons (three comparisons) with the Holm correction. ***$p$ value <0.001. Exact $p$ values and statistics are presented in Supplementary Information section "Statistics for physiology in exploration scenarios", Tables S6–7. Correlation between the *XGBoost* model's prediction and the integrated HRV index (iHRV) for the test set **c** and train + test set **d**. **e** SHAP values for the trained model, calculated for each subject. The nine most discriminating features (from the 18 used by the model) were the minimum distance to the corners of the empty room [Corner dist (er)], vertical acceleration in the elevated alley [Vert acc (ea)], ratio between time in the center and time in the periphery of the empty room [Ratio time (er)], time spent on the narrowest ledge of the elevated alley [Time narrow (ea)], movement focus in the empty room [Focus (er)], variability in the stride speed in the empty room [Stride speed var (er)], number of head scans while walking in the empty room [Walking head scan (er)], time spent on the starting board in the elevated alley [Time board (ea)], maximal longitudinal distance reached in the elevated alley [Distance long (ea)].

set contained participants who presented low (STAI-T < 35) and moderately high (STAI-T > 45) scores to ensure that we had a good representation of participants' iHRV in each of the two data sets, and to prevent biasing HRV prediction on trait anxiety. The test/replication set contained participants with moderate (35 > STAI-T > 45) scores. To fit a linear response function to iHRV, we used a method based on extreme gradient boosting trees (*XGBoost*)[62]. The hyperparameters of the *XGBoost* algorithm were tuned with Bayesian optimization using the Python package *hyperopt*[63] (for more details on *XGBoost* and hyperparameter search, see Supplementary Information).

Following training, our model's predictions also showed high correlations with the iHRV values from (i) the training/discovery data set ($r = 0.91$, $p < 0.001$; for a 10-fold validation performance of the model on the training data set, see Supplementary Fig. 3); (ii) the test/replication data set ($r = 0.54$, $p < 0.001$; Fig. 2c); and (iii) the training and test data sets together ($r = 0.72$, $p < 0.001$; Fig. 2d). These results confirmed our expectation that rich

behavioral data from individuals' reactivity to novelty challenges would predict variation in a physiological measurement of stress vulnerability, such as vagal-mediated HRV.

**Understanding key behavioral features for model prediction.** To understand the impact of each behavioral feature in the model's iHRV prediction, we examined the Shapley additive explanation (SHAP) values[64,65] for each individual feature (Fig. 2e). The most discriminating features included positioning and gait parameters, with the largest contribution from the former category, including minimum distance to the corners of the empty room [*Corner dist (er)*], vertical acceleration in the elevated alley [*Vert acc (ea)*], and ratio between time in the center and time in the periphery of the empty room. Specifically, the model predicted a lower iHRV (i.e., a larger parasympathetic withdrawal value) for subjects who stayed closer to the corners of the empty room and had higher vertical acceleration in the elevated alley. In

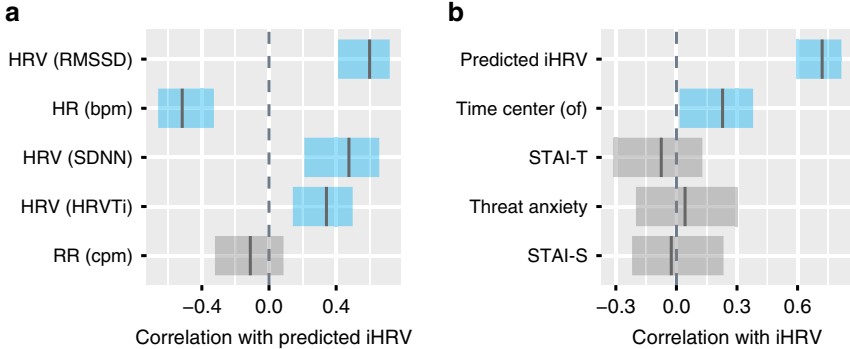

**Fig. 3 Correlation plots. a** Correlation between the model's prediction of PC1 (i.e., iHRV) for the entire data set with each of the cardiorespiratory variables used to create PC1. **b** Comparison between PC1 and the model's prediction and the correlation between PC1 and other possible explanatory variables. *Time center (er)*—time in the center of the empty room; *STAI-T* - self-reported trait anxiety; *Threat anxiety*—self-reported anxiety felt during the persistent threat scenario; *STAI-S*—self-reported state anxiety. Black line in the center of shaded area are Pearson correlation coefficients; shaded area, 95% confidence intervals for the correlation coefficients, corrected for multiple comparisons in each panel; blue shaded area, significant correlation corrected for multiple comparisons in each panel.

addition, the model predicted that these subjects would spend less time in the center of the empty room and on the narrowest ledge of the elevated alley; they would also show a more focused exploratory trajectory, less variability in stride speed and fewer head scans while walking in the empty room.

As expected, the model's predictions were positively correlated with HRV computed with the formulas, RMSSD ($r = 0.60$, $p < 0.001$), SDNN ($r = 0.48$, $p < 0.001$) and HRVTi ($r = 0.34$, $p < 0.001$), and negatively correlated with HR ($r = -0.52$, $p < 0.001$) (Fig. 3a). Importantly, iHRV did not correlate with RR ($r = -0.11$, $p = 0.199$). This suggests that the computation of an iHRV for model training was successful in providing a surrogate measure of HRV for model learning that allows good generalization.

The correlation between the model's prediction and iHRV ($r = 0.72$, $p < 0.001$) was markedly higher than that of other possible explanatory variables for parasympathetic influence (Fig. 3b). Self-reported trait anxiety, state anxiety, and anxiety felt during scenario 3 were not correlated with iHRV ($-0.1 <$ all $r$s $< 0.1$, all $p$s $> 0.386$). Time spent in the center of the empty room [*Time in center (er)*, see Supplementary Information section "Behavioral feature description", Table S2] correlated significantly ($r = 0.23$, $p = 0.008$) but with a low coefficient compared with the model's prediction.

**Model generalization to other stressful challenges**. To further validate our model's prognostic power and explore whether it would generalize to other situations, we compared its predictions with the cardiac response of a generalization set comprising 107 out of the initial 135 participants (age = 20.48 ± 2.19 years) who were exposed to another stressful challenge in VR for 10 min (Fig. 4a, b). As iHRV was computed to train and test the model, the model's predictions were directly compared with markers of HRV (the RMSSD) and HR. We found significant correlations between the predictions and HRV (RMSSD) and HR ($r = 0.38$, $p < 0.001$ and $r = -0.40$, $p < 0.001$, respectively; see Fig. 4c, d). Furthermore, the predictions were also significantly correlated with HRV (RMSD) and HR measured from pulse data—the pulse rate variability ($r = 2.4$, $p = 0.016$; Fig. 4e) and pulse rate (PR; $r = -0.32$, $p < 0.001$; Fig. 4f) obtained during a sustained anticipatory anxiety paradigm (elicited by shock anticipation) performed on a separate day by subjects in the generalization set. See Supplementary Information for further details. Altogether, these results support the view that the model predictions can generalize to other stressful challenges and anticipatory anxiety situations.

## Discussion

In this study, we demonstrate the strong predictive value of behavior to forecast individual differences in HRV responsiveness to stress. Specifically, we show that high-dimensional locomotor data obtained during novelty exploration can be used to predict interindividual differences in the parasympathetic index of HRV (iHRV) during stress exposure. Notably, our model was validated in a different cohort and generalized across different stressful situations.

Novelty exploration is one of the most frequently studied behavioral manifestations in laboratory animals and is a remarkable way to reveal the complexity of behavior and its links to psychopathology[56,66]. Following a reverse-translational approach, we designed two VR scenarios for human behavioral phenotyping (the empty room and the elevated alley) inspired by classical rodent exploration tests; i.e., the open field and the elevated plus maze, respectively. Both rodent tests exploit the tradeoff between exploratory tendencies and natural aversion for open spaces. Although they were originally designed to assess anxiety-like responses, ambulatory parameters can also provide information regarding activity and decision-making processes[67,68]. Notably, segregation of animals according to anxiety-like behaviors and novelty reactivity in these tests identified differential susceptibility to the development of stress-induced depressive behaviors[32,33,35,69].

Of a number of variables collected from each participant's locomotor responses, the most discriminating variables contributing to our iHRV predictive model include features (e.g., minimum distance to the corners of the empty room, ratio between time in the center and time in the periphery of the empty room, and time spent on the narrowest ledge of the elevated alley) that are classically interpreted as anxiety-like behaviors in rodents. These features relate to the tradeoff between boldly exploring all areas, including more anxiogenic ones, versus predominantly staying in more protected areas. On the high-anxiety side of the spectrum, these variables (e.g., thigmotactic movements around the perimeter of the empty room or staying on the starting platform in the elevated alley) define variation in behaviors related to establishing a "home base" from which spatial exploration may occur[70]. Other variables (e.g., vertical acceleration in the elevated alley, time spent on the starting board in the elevated alley, and number of head scans while walking in the empty room) also relate to features defined as anxiety-like behaviors in rodents[67,68]. In addition, other discriminating variables were indicative of decision-making processes (e.g.,

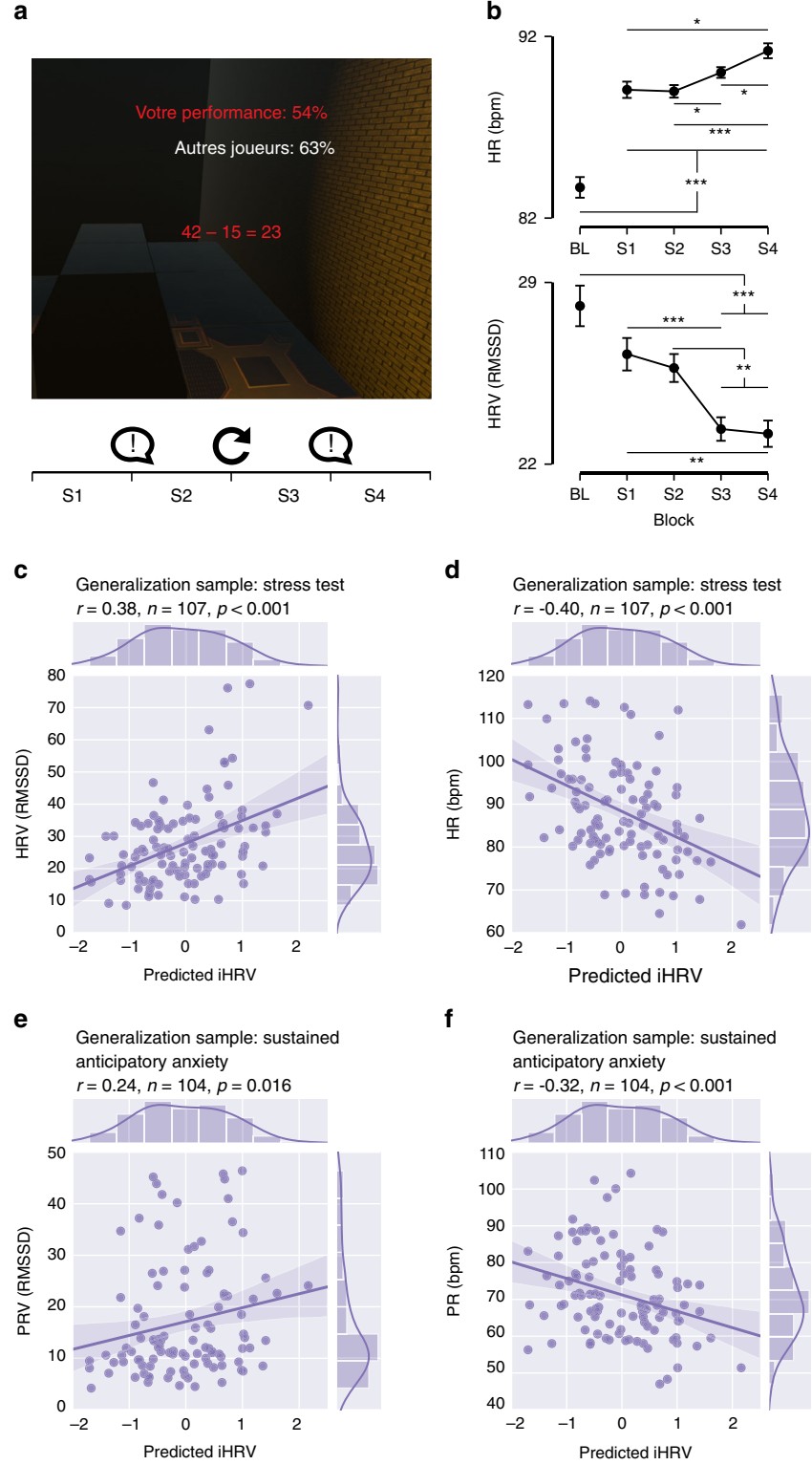

movement focus in the empty room and variability in the stride speed in the empty room) and activity (e.g., maximal longitudinal distance reached in the elevated alley). Although anxiety disorders are frequently accompanied by alterations in HRV[71,72], in our nonclinical population, none of the trait or state anxiety or threat anxiety measures predicted iHRV. Self-reported anxiety measurements, as obtained with the STAI inventory, typically explain a lower percentage of the variance in stress reactivity (see, e.g.,[73]) than the one achieved by our machine learning model. Therefore,

the lack of correlation between anxiety measures and our model predictions supports the view that our model has achieved a high level of precision in capturing specific physiological reactivity (as opposed to other non-related behavioral predispositions), further supporting its specificity in predicting stress-induced iHRV/cardiovascular responses.

A few studies have used human versions of the open field and elevated plus maze tests. They were based in real-life contexts[70,74,75], computer-based contexts[70] or mixed reality

**Fig. 4 Generalization of model prediction of iHRV to other stressful or anxiogenic situations and a generalization subsample. a** Schematic of a virtual reality stress test involving mental arithmetic and challenging contextual navigation. This scenario lasts 10 min and is split into 2.5-minute segments (S1–S4). After segments S1, S2, and S3, participants are presented with a prerecorded voice asking them to perform better. **b** Physiological (HR, HRV) responses to the stress induction task. During segment BL, baseline recordings are taken. $N = 99$ participants examined over five blocks of 2.5 min (Bl, S1, S2, S3, and S4). Data are presented as mean values ±SEM (vertical lines). A repeated measures analysis of variance (rm-ANOVA) was performed for HR and to HRV. Post hoc tests were performed with two-sided paired $t$ tests, with $p$ values corrected for multiple comparisons (10 comparisons) with the Holm correction. *$p$ value < 0.05, **$p$ value < 0.01, ***$p$ < 0.001. Exact $p$ values and statistics are presented in Supplementary Information section "Statistics for physiology in Fig. 4b", Tables S13–14. **c** Spearman correlations (two-tailed $p = 5.258e-5$) between the model predictions for iHRV and HRV for the sample of participants who underwent the stress task. **d** Spearman correlation (two-tailed $p = 2.523e-5$) between the model predictions for iHRV and HR for the sample of participants who underwent the stress task. **e** Spearman correlation (two-tailed $p = 0.016$) between the model predictions for iHRV and pulse rate variability (PRV) for the sample of participants in the fear anticipation condition. **f** Spearman correlation (two-tailed $p = 9.739e-4$) between the predicted value of iHRV and pulse rate (PR) for the sample of participants in the fear anticipation condition. 95% confidence interval for the regression lines drawn using translucent bands.

combining VR and real world elements[76]. Although these studies confirmed the validity of the tests in identifying individual differences in anxiety, including clinical cases, their approaches differ from ours in several ways. First, the goals are different. We used VR phenotyping tests not to define personality traits or states as in previous studies, but to gather meaningful locomotor data to feed a machine learning model that could predict physiological vulnerability to stress. Second, the methods are different. As opposed to computer-based approaches with low immersive capacity or real-life mazes and scenarios that lack reproducibility power and are time consuming, our VR resource probes individuals' novelty reactivity at high efficiency (i.e., the space requirements for our VR scenarios are of a standard room size, and the sampling times are short). Third, and most importantly, the approaches are different. Previous studies have relied on a restricted number of classical variables, remaining confirmatory as to the enhanced prevalence of thigmotactic behavior and avoidance of open spaces in highly anxious individuals. However, limited sets of behavioral readouts can result in noisy and inconsistent outcomes. Instead, high-throughput behavioral readouts are desirable, as they provide high dimensionality, which can then be explored by mathematical frameworks to find the key parameters whose consideration optimizes the identification of interindividual differences[77]. In fact, the prognostic capacity of the the model using information from multiple behavioral features outperforms a single behavioral feature such as time spent in the center of the empty room, which is the prototypical parameter relating to anxiety in human open-field tests[74,75].

Thus, a key advantage of our model is that instead of relying on individual variables to discriminate behavior, it integrates behavior across a number of highly discriminating variables, each of them providing a different weight to the decision-making embedded in the prediction process. Similar approaches have been successfully applied in recent studies on laboratory animals to link specific behaviors and traits from high-density locomotor data to neural substrates[77–79].

Most biomarkers of stress vulnerability that research currently considers rely on biological measurements that frequently involve invasive approaches, such as blood extraction, or low-throughput neuroimaging technologies[37,80–82] that deter from their systematic use. Although HRV measures are easier to collect and are emerging as particularly relevant[36–38], establishing when and how phasic HRV predicts physiological vulnerability to stress requires standardized and validated conditions. Resting HRV has been associated with health, but ensuring its reliability requires long recordings[49,58]. Importantly, our model's predictive capacity outperformed that of individual baseline HRV values obtained in the empty room.

Although wearable and other digital technologies can provide valuable real-life cardiovascular data, the interpretation of these data is not straightforward, particularly owing to the overwhelming diversity of human experiences and constraints. Although HRV can be computed from ECG or pulse data, the reliability of these measures when extracted from wearable device data is still unconfirmed for brief short non-rest recordings[83,84], which is typically how stress occurs. Meaningfully, short-term cardiovascular reactivity, including HRV, elicited by emotional challenges in the laboratory has been shown to provide a fairly accurate representation of HRV responses to real-life stressors[49] and to outperform the predictive capacity of resting assessments in studying disease development[6,46–48]. Thus, our approach using controlled experimental VR conditions represents a major step forward in identifying key behavioral features. Accordingly, it can be adapted in the future to develop diagnostic tools for biomarker discovery and guide implementation in predictive analyses of physiological stress susceptibility on data from digital technologies. In addition, our model allows incremental training by iterative training with new observations, which allows us to encompass a larger representation of different traits. Furthermore, approaches based on transfer learning can also enable the adaptation of our model to other target domains such as clinical settings or even, in a similar fashion to mapping biological relationships from mouse to human[85], to facilitate translational approaches mapping animal behavioral/physiological interplay to human models or vice-versa.

On the other hand, as compared with non-VR based laboratory stress challenges, such as the trier stress test (TSST[86]) or the socially evaluated cold pressor test (SECPT[87]), our IVEs constitute a more standardized procedure allowing for more reproducible and controlled conditions across subjects and laboratories. In the future, it will be important to benchmark the capacity of our IVEs to both, trigger physiological responses and predict stress vulnerability against procedures such as the TSST or SECPT.

Importantly, our study included only males given that our prediction linking behavioral and physiological responses to stress vulnerability was inspired in previous studies involving male rodents[33,88] and, therefore, future work is warranted to assess the validity of our model to predict iHRV responses in females. In addition, although our results indicate a potential high predictive capacity of our model for later psychopathology than existing tools/questionnaires, it will be important to benchmark it in both longitudinal studies and clinical populations.

In conclusion, the present study emphasizes the power of behavior to predict HRV responsiveness to a subsequent stressful challenge. Thus, in addition to highlighting behavior as a potential stress vulnerability marker, our study contributes a relevant approach to develop diagnostic tests based on VR immersion and machine learning modeling. In the future, our approach could be implemented to identify stress-susceptible

individuals in longitudinal cohorts using recent advances in clinical digital phenotyping[19,21]. Considering how readily locomotion features can be extracted from mobile phones[89] or in-home radiofrequency tracking[90], we hope our work will pave the way for new technological implementations aimed at far-reaching personalized prevention of the negative outcomes of stress.

## Methods

**Participants**. These results are part of a larger study aimed at investigating behavioral and physiological predictors of aggressive behaviors. Here, in order to specifically addressed the main question of our study, we only analyzed a subset of all collected variables, with respect to locomotor features and parasympathetic responses. One hundred thirty-five (135) male participants between the ages of 18 and 38 (age: $20.58 \pm 2.05$ years) were recruited. Participants reported that they had not been diagnosed with psychiatric disorders and did not use psychotropic medication. The study was approved by the Cantonal Ethics Committee of Vaud, Switzerland. Participants were asked to refrain from eating or drinking (except water) one hour before the experiment took place. Informed written consent was obtained from all participants. Days before the experiment, participants were asked to complete the Spielberger Trait Anxiety Inventory (STAI-T, form Y[91]) and other personality questionnaires. After the experiment participants also completed the Igroup Presence Questionnaire[92]. For further information about participants and recruitment, see Supplementary Information.

For the purposes of model learning and testing, we split these 135 participants into training/discovery ($N = 66$ subjects, age: $21.00 \pm 1.93$ years) and testing/replication ($N = 69$ subjects, age: $20.19 \pm 2.10$ years) data sets. The training/discovery set contained participants who presented low (STAI-T < 35) and moderately high (STAI-T > 45) scores. The test/replication set contained participants with moderate (35 > STAI-T > 45) scores. Furthermore, 107 out of the 135 participants (age: $20.48 \pm 2.19$ years) additionally participated in a stressful task and a sustained anticipatory anxiety task; this sample served as the generalization sample.

**Experimental procedure**. Participants were asked to explore three different virtual scenarios for 90 s each (see Figs. 1a, b and 2a), separated by two short transitions. At the beginning of each trial, participants were instructed to explore the current scenario, all starting from the same initial position. After participants agreed to continue, 90 s were left for exploration. At the end of the 90-s period, participants were instructed to return to the starting point. The subgroup of participants who participated in the stress task did so after the exploration scenarios, which were also in VR. Several days prior to the VR tasks, the same subgroup of participants was invited to perform a sustained anticipatory anxiety paradigm.

Scenario 1: Empty room: In the first trial, the participants started at the edge of an empty room (Fig. 1a) on top of a small red step, facing one of the walls. Participants were briefly instructed to explore the empty room. The room's dimensions were the same as our room's physical dimensions; hence, if a participant touched the virtual walls, he would feel the real walls. After 90 s, participants were instructed to return to the red step and wait for further experiments. The scene faded to black until the next scenario was loaded. After a brief transition, the new scene was a street in a virtual city. After being briefly instructed not to move away from the red step, the participants began being lifted towards the elevated alley in Scenario 2.

Scenario 2: Elevated alley: In this trial, participants were exposed to an elevated virtual alley several meters above the ground in a virtual city. The alley was wider on the starting side (3.50 m) and became narrower on the opposite side (0.50 m). At the edges next to the starting point, there were two walls so that participants were not exposed to the height at the starting point. Participants were asked to explore the alley, and after 90 s, they were asked to return to the red step. After a transition, participants were standing on top of the red step; the elevated alley was no longer present. After participants listened to a brief information statement and agreed to continue, the step started descending with accelerated movement until it reached the ground level. The scene faded to black until the next scenario was loaded.

Scenario 3: Persistent threat: The scene faded into a completely dark room. In this trial, participants were asked to explore a darkened maze corridor. The hand controller served a flashlight and as the only source of light in this scenario. Four human-like static Fig.s were placed in corner areas, and three bursts of white noise were delivered to the participant's headphones at $t = 20$ s, $t = 40$ s and $t = 60$ s. After 90 s, participants were instructed to return to the red step to end the exploration experiment.

Stress test: This test exposed participants to an uncontrollable social-evaluative task and timed problem solving with negative feedback in a challenge in VR (Fig. 4a). The design of this test followed the recommendations presented in ref. [93] for successful stress induction and activation of the HPA axis by posing a threat to central goals. The task comprised a motivated performance task, relative uncontrollability of task outcomes, and the presence of social evaluation. Similar to the Montreal Imaging Stress Test[94], participants had to solve quick arithmetic tasks, and their responses were recorded as correct or incorrect. Incorrect responses caused a tile on the floor to break and disappear, which could cause a fall if it were

stepped on. If participants fell, they would fall into a new room with full tiles on the floor. Before the test started, participants had 3 min to read the instructions, which informed them about the ensuing task and objectives (get as many correct responses as possible and avoid falling) and indicated that they would be recorded by video. Two minutes of training preceded the 10-minute session. The test titrated participants' performance to be below a supposed "average of the other participants" (63%) by reducing the response time limit. This supposed average was shown above the participant's score, which turned red or green if it was below or above the hypothetic average, respectively. There was a 5% chance that correct responses were recorded as incorrect to increase the feeling of uncontrollability. A prerecorded voice asked the participants to perform better 2.5 and 7.5 min after the start of the test. Five minutes after the test started, the same prerecorded voice informed the participants that their performance was not good enough and that the test would restart.

Baseline physiology was also recorded inside VR in equivalent conditions but without the stressful elements. Participants were immersed in a nature setting, and no video recording was mentioned. In addition, for the first 7.5 min, they were prompted to respond to similar arithmetic tasks as in the stress challenge, but the tasks were easier and had a longer time limit. During the last 2.5 min, participants were told they could control the passage of time in the virtual world by pressing the controller's trigger. Baselines were taken during this part.

Sustained anticipatory anxiety paradigm: This paradigm was based on a classical differential delayed fear conditioning paradigm[95] during the 2 min of the habituation phase (see Supplementary Information for further details). Several days prior to the tasks in VR, participants were invited to our laboratory to participate in the sustained anticipatory anxiety paradigm. We used Psychlab Contact Precision Instruments SHK1 to present shocks. Shocks were administered to the top of the right wrist using 2 Ag/AgCl electrodes (6 mm). Shock intensity was determined individually during a workup procedure performed before the sustained anticipatory anxiety paradigm[96]. Pulse and skin conductance were measured from participants' left hand, which was resting on a cushion on the table. Before the start of the paradigm, participants were informed they would be presented with two images and that electric shocks would be delivered occasionally. During the first two minutes (habituation phase), no electric shocks were actually delivered, and only the images were presented. As participants had been informed that electric shocks could be delivered, we used this block to induce a sustained anxious state[97]. Four subjects were excluded due to missing data in this experiment.

**Experimental setup**. VR was performed using a commercial VR system developed by HTC and Valve (HTC Vive). The head-mounted display (HMD) presents stereoscopic images of the virtual scene to the participant with a resolution of $1080 \times 1200$ pixels per eye and a refresh rate of 90 Hertz. The tracking relies on two *lighthouse* stations sweeping structured light lasers into the testing room. The HMD uses several sensors (laser position sensors, microelectromechanical sensors, gyroscope and accelerometer) to reliable infer their position and orientation in 3D space in real-time and with sub-millimeter precision. The different 3D scenes were rendered in Unity3D (www.unity3d.com) running on a computer dedicated for VR and motion capture, equipped with a Core i7 CPU clocked at 4.0 GHz, 16GB of main memory and a GeForce GTX TITAN X graphics card. 3D scenes were modeled in Blender (https://www.blender.org) and imported into Unity3D. The experimental logic was programmed in C# within Unity3D. The app-to-display latency for the HTC Vive running Unity3D apps in a computer with a similar configuration to the one used in this experiment has been determined to be on average 31.33 ms with a standard deviation of 1.41 ms[98].

The dimensions of the testing room are 3.50 m (width), 6.00 m (length), and 3.50 m (height). Within this range, participants could move around freely. We ensured that participants could never see the physical room to increase immersion and the sense of novelty when exploring the virtual scenarios.

Motion capture was performed using the MVN XSENS Awinda system on participants' lower body (pelvis, legs, and feet sensors) and recorded using MVN Studio Animate Pro. The gait and center of mass displacement were computed from the motion capture data.

Experiments in VR were performed with a wireless physiology recording system (Biopac Bionomadix) recording data at a 1000 Hz sampling rate with AcqKnowledge Data Acquisition and Analysis Software 5.0. We recorded respiration and electrocardiogram (ECG) data. ECG was decimated to 500 Hz and breathing to 100 Hz using the MATLAB function decimate with the default FIR filtering prior to down sampling. For the sustained anticipatory anxiety paradigm, we recorded pulse and skin conductance with a wired Biopac MP150 unit collected at 250 Hz with AcqKnowledge Data Acquisition and Analysis Software 5.0.

HR and PR were computed from AcqKnowledge's default functions. ECG R peak detection was performed with the Pan Tompkins algorithm. The resulting RR time series was filtered with a 3-standard-deviation filter and visually inspected for artifacts. Linear interpolation was used for the identified artifactual beats. HRV formulas were computed from the resulting RR time series.

RR was computed using the MATLAB function *findpeaks* after filtering the respiration signal with a bandpass FIR of 0.17–0.73 Hz. We identified peaks above at least one standard deviation (robustified) of the filtered respiration time series, separated by at least 0.8 s and with a minimum width of 0.4 s.

For the VR exploration scenarios, cardiorespiratory variables were computed for the entire length of the 90 s intervals of exploration. For the stress task and for the sustained anticipatory anxiety paradigm, we computed rate and variability variables for the entire length of the tasks (10 and 2 min, respectively).

**HRV analysis**. HRV analysis was performed on the filtered R to R time series. Among the vast number of available HRV formulas, we decided to use the most widely used time-domain formulas for parasympathetic activity[60]: RMSSD, SDNN, and HRVTi. Details concerning these formulas can be found in[58]. Further information regarding HRV can be found in the Supplementary Information.

RMSSD is highly correlated with high frequency power[45], another widely used measure of the parasympathetic (vagal) nervous system[58,99], which is known to be influenced by respiration[99]. Controlled breathing has been shown to have an effect on RMSSD compared with uncontrolled breathing[100]. However, to obtain a more generalizable parasympathetic metric that was resilient to breathing, we performed PCA on the HR, HRV (RMSSD, SDNN, and HRVTi) and RR variables and chose the principal component that loaded onto the HRV and HR variables but not onto RR.

**Behavioral measurements**. Behavior was assessed via position tracking and motion capture. More specifically, with position tracking, we computed several features regarding where participants positioned themselves in the virtual space, velocity and acceleration of movement, and other trajectory features obtained from[101]. With motion capture, we computed gait variables. A detailed list of all behavioral features can be obtained in the Supplementary Information.

**Feature selection**. Feature selection is an important step when building predictive models. Removing redundant features can simplify the generated model, prevent overfitting, and enhance the generalization ability. In this study, feature selection was performed on the training set. Zero variation features were removed, as were features with a Spearman correlation coefficient below 0.1 with the parasympathetic PC during the persistent threat scenario. The resulting features were separated into three different tables: gait, movement burst and position tracking. For each table, cross-validated Lasso regularization of the generalized linear model with MATLAB's *lassoGLM* function was used to predict the iHRV during the persistent threat scenario. The resulting unshrunk variables for each table were then taken together as the selected feature for statistical model learning.

**Extreme gradient boosting trees (XGBoost)**. *XGBoost*[62] is a new implementation of the gradient tree boosting technique that has been tested in different data sets; it achieves high accuracy and requires much less computation time than deep neural nets[102]. It is known for obtaining winning solutions in various data competitions. It is authors reported that "among the 29 challenge-winning solutions published on Kaggle's blog during 2015, 17 winning solutions used XGBoost."[62]. *XGBoost* has also been applied in the medical field[103–105]. Here, we used *XGBoost* to predict iHRV during the persistent threat scenario with the set of selected behavioral features. We used the *XGBRegressor* function from the Python *XGBoost* package to fit our model. There are several adjustable hyperparameters in *XGBoost*. In this study, the step size shrinkage (eta), maximum depth of tree (max.depth), minimum sum of instance weight (min.child.weight), and maximum number of iterations (nrounds) were optimized with Bayesian optimization[106] using the Python package *hyperopt*. The performance of our model's predictions was evaluated by correlating the predicted with the true value of the parasympathetic PC on a holdout sample (not used in model learning). Train and test data sets were loaded into data frames using the *pandas* Python package, processed with *datacleaner* Python package and model evaluation metrics computed with the Python package *numpy*. SHAP values were computed with the Python package *shap*.

**Statistics**. Statistical analyses were performed using MATLAB, R, or JASP. All tests were two-sided and normality of the underlying data distributions was assumed when parametric tests were used (for example, analysis of variance in Supplementary Information). For correlations, normality was tested with the Shapiro–Wilk normality test and Spearman correlations were used, instead of Pearson, if data were not normal. All correction for multiple comparisons were performed using the Holm procedure. Correlation confidence intervals are depicted in the corresponding Figures.

**Reporting summary**. Further information on research design is available in the Nature Research Reporting Summary linked to this article.

## Data availability

The data that support the findings of this study are available from the corresponding author upon request (note that the data analyzed here is part of a longer undergoing study with a different aim and not yet published; as soon as the remaining parts of the data/study are completed, we will make our data available in a public repository). Source data are provided with this paper.

## Code availability

The code used to train and test the classifier in Python will be made available upon request. Likewise, the feature selection, gait analysis, and HRV implemented in MATLAB will be made available upon request (note that the data analyzed here is part of a longer undergoing study with a different aim and not yet published; as soon as the remaining parts of the data/study are completed, we will make our code available in a public repository).

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

## Acknowledgements

This work was supported by grants from The Oak Foundation, the Swiss National Science Foundation (SNSF; NCCR Synapsy grant nos. 158776 and 185897; Sinergia no. 183564;) and intramural funding from the EPFL to C.S. and an SNSF SPARK grant to J.R. The funders had no role in study design, data collection and analysis, decision to publish or preparation of the manuscript.

## Author contributions

J.R., E.S., S.S., and C.S. designed the experiments; J.R., E.S., S.S., and N.M. performed experiments; J.R. and E.S. analyzed the physiological data; J.R. analyzed the motion capture data and applied machine learning analyses; J.R. and C.S. wrote the manuscript; C.S. supervised and supported the project.

## Competing interests

The authors declare no competing interests.
