## [Peer Review File · Nature Communications]

Reviewer #1 (Remarks to the Author):

Rodrigues and colleagues tested the predictive value of newly developed virtual reality (VR) tools for subsequent heart rate variability (HRV) responses to acute stress. The authors report strong correlations between high-dimensional locomotor responses in specific VR environments with stress-related HRV.

This study represents an excellent example of how the translation of findings from the rodent literature (here, the open field and elevated plus maze tasks) to humans might work. The findings are novel and highly relevant for clinical contexts as they may pave the way to the development of non-invasive predictors for cardiovascular or mental disorders. A further strength of this study is the link between VR-based predictors and anxiety responses in a fear conditioning protocol tested on a separate day. I have no doubts that this study would make a unique contribution to the existing literature. I have only relatively minor comments.

1) I assume the authors collected also some baseline measurements of HRV. Did these baseline measures of HRV already predict HRV in the stressful situation (and potentially also pulse variability in the fear conditioning task)? If so, did the locomotor responses obtained in VR predict stress-related HRV beyond baseline HRV?

2) It is stated in the methods that only male participants were included. I assume this was motivated based on the rodent literature, which focusses also mainly on males. Nevertheless, the focus on males should be addressed as a potential limitation in the discussion.

3) The authors focus exclusively on HRV, which is well justified. I was still wondering about the predictive value of the locomotor responses in the VR for other parameters that may be linked to psychopathologies. For instance, did the authors measure cortisol responses to the Montreal Imaging Stress test? If so, were these also predicted by the locomotor responses? More generally, would the authors expect that the obtained locomotor responses predict other arousal/stress measures beyond HRV?

4) I was surprised that the locomotor responses were not at all correlated with trait or state anxiety measures. How do the authors explain the absence of such correlations? What are the implications of the lack of such correlations for the conclusions of the authors and for the potential use of the locomotor responses as a predictive tool for later psychopathology?

5) In the methods, it was not entirely clear to me how participants were assigned to the training or test data sets. Was this solely based on the mentioned anxiety related criteria? May the definition of the training vs. test samples based on anxiety-related criteria have affected the potential predictive value for state and trait anxiety?

6) Related to the previous comment: was the assignment of participants to the training vs test data sets varied and did the authors use some form of permutation test to test the robustness of the observed effects? Alternatively, one might also think about a variant of the 'leave-one-out' methodology?

7) Finally, the authors suggest that their VR protocol might be used as a predictive tool for cardiovascular diseases or mental disorders and I completely agree that this protocol may be helpful in this respect. Perhaps the authors could still elaborate in the discussion on the (dis)advantages (pros and cons) of their protocol relative to simply using another standardized stress protocol such as the TSST (PMID: 8255414) or the SECPT (PMID: 18403130).

Reviewer #2 (Remarks to the Author):

Parsing heterogeneity in stress responsivity is becoming an increasingly important research goal. This paper therefore represents an important contribution as it presents interesting data towards accurate characterization of resilience versus sensitivity to stress with implications for the implementation of neuroscientific findings into clinical care. In order to be suitable for publication

in Nature Communications, however, some issues need to be clarified. The comments below are intended to assist the authors in sharpening a future revision of their paper.

-In the Introduction, the sentence: "Stress-related cardiovascular reactivity values elicited in the laboratory setting have been shown to outperform the predictive capacity of resting assessments in studying disease development and have been found to be stable personal characteristics that are fairly consistent across time and between stressors" is not completely correct. In the field of cardiovascular disease, the so-called reactivity hypothesis has failed to fulfill its promise leading to inconsistent findings (see for example the Whitehall II Study; Carroll et al., 2001). The authors should acknowledge that cardiovascular responses to the brief and unecological stressors that are commonly used in the lab have serious limits that have been overcome by more expensive and time-consuming ecological ambulatory studies (e.g., studies by Pickering, Gerin, Shapiro, etc.). This is in my opinion, a major advantage of the proposed study which makes an attempt to take ecological stressful situations into the lab. In fact, the authors do not use cardiovascular reactivity as a predictor, but a complex combination of behavioral features (i.e. 172 behavioral features to start with). Please revise the Introduction to make this point clearer.

- "To define stress vulnerability, we focused on heart rate variability (HRV) a surrogate index of cardiac vagal break that yields information about autonomic flexibility". Could the authors please better introduce the reasons why they have chosen this particular index as a measure of stress vulnerability? There are good reasons for this choice, but in my opinion this rationale should be better explained.

- In the Introduction "Low HRV values are associated with increased risk of ..." the authors should distinguish between tonic and phasic HRV, which provide different information and have different prognostic value. This distinction should be made clearer throughout the manuscript.

-When they introduce the behavioral features ("The most discriminating features included positioning and gait parameters"), the authors should briefly describe the meaning of such features to help the reader's understanding. These are mentioned in the Discussion ("e.g., minimum distance to the corners of the empty room, ratio between time in the center and time in the periphery of the empty room, and time spent on the narrowest ledge of the elevated alley") but in my opinion it would be useful to move this section earlier in the text.

-I may have missed it, but is there an association between the identified behavioral features and self-reported trait/state anxiety?

-The sentence "Although anxiety disorders are frequently accompanied by alterations in HRV, in our nonclinical population, none of the trait or state anxiety or threat anxiety measures predicted iHRV. This finding further supports the robustness of our model in predicting cardiovascular stress vulnerability" is not clear to me. Why this further supports the robustness of the model?

-Participants section (Method): were participants asked to refrain from caffeine, alcohol, strenuous exercise, etc. before the experimental session?

-Participants are all males. Please comment on this choice which prevents generalization of results to the general population.

Minor:

Page 3, line 110: "responses progressively habituated during immersion VR in each scenario". I think that there is a typo and that "VR" should be deleted.

Reviewer #3 (Remarks to the Author):

In this paper, the authors have developed three VR scenarios with locomotion tasks for phenotyping inspired by the animal literature: (i) open room, (ii) elevated alley, and (iii) dark maze corridor and computed individuals' HRV responses to stress. With the help of machine learning methods, they show the predictive power of high-dimensional locomotor responses to

predict HRV responsiveness during stress exposure. Their model's predictions generalized to a different cohort and to other stressful challenges and outperform other stress prediction instruments, such as anxiety questionnaires.

To my knowledge, no study has demonstrated the prognostic value of behavioral data as a predictor of stress vulnerability using objective (i.e., not self-reported) stress measurements. Hence, the work provides a significant research contribution to the field of locomotion research and prediction of cardiovascular responsiveness to subsequent stressful situations.

The paper addresses an important topic, but there are also several issues.

Overall, I think the findings are not very surprising. In previous work stress-related cardiovascular reactivity values elicited in the laboratory setting have been shown to outperform the predictive capacity of resting assessments in studying disease development. The authors aimed to replicate this by developing immersive virtual environments (IVEs) to predict physiological stress susceptibility.

While the ML algorithms perform greatly to the training data (as expected), the correlation for the test/replication dataset is rather low. The authors discuss in more details the reasons for the rather low correlation and discuss the findings from the previous work on stress-related cardiovascular reactivity values elicited in the laboratory setting.

Next, the authors argue that it is important to develop tools that identify predictive markers of individual differences in HRV responsiveness without exposing subjects to high stress. While I can follow this argument in principle, it would be good if authors could discuss the motivation in a bit more detail.

Furthermore, there are several missing details about the experimental setup and procedure, which need to be clarified in the revision. For instance, which HMD was used? HTC Vive Pro / Eye? Which computer was used for rendering? How was the tracking implemented? Which VR system (Unity3D?) was used to render the VE? How much end-to-end latency is induced by the system. The authors describe that subjects could never see the physical room to increase immersion and the sense of novelty. I think the authors mean a sense of presence.

Also, the machine learning algorithms is rarely explained. Was this based on a CNN? Finally, I think the authors should explain in more detail why they decided on the provided scenarios, which only partly mimic the setups from the animal literature, and which could have been chosen very differently. Were the scenarios randomized between subjects? If not, why?

As a minor remark there is some inconsistency throughout the paper, e.g., "head-mounted display" vs. "head mounted display", they introduced the acronym HMD, but never used it etc.

To summarise, I think the paper lacks too many details and discussions of the results, which need to be added before the paper can be accepted. However, the findings appear to be a significant research contribution and very interesting, hence, I would recommend a major revision.

Reviewer #4 (Remarks to the Author):

This paper is about predicting physiological vulnerability in response to stress. It builds on observations from animal studies where it is shown that behaviors inducing anxiety and reactions to novelty can subsequently explain the induction of susceptibility to depressive behaviors.

The novelty in the work presented here is to build on it for human subjects, but carefully design virtual reality environments via which stressful stimuli may be induced, the response to such stress quantified in a high dimensional space by features extracted from wearable sensor measurements and subsequently predict behavior in a previously unseen but a more severe task. The work is imaginative in setting up the VR environments, using heart rate variability as proxy for induced physiological state and careful in decoupling any systematic confounding effects (e.g. respiration). The manuscript is clearly written and easy to follow. Figure captions are sufficiently detailed and informative. As such, I feel positive towards recommending the work for publication. However, I have a major reservation that needs to be addressed before publication, which has to

do with the results obtained via machine learning, clearly apparent in Fig. 2(c) and (d) as a huge disparity between predictions made on the training set and the test set. This will usually be seen as an indication of overtraining in usual machine learning problems, where the training and test sets arise from the same joint distribution (of inputs and responses). In the problem considered here, however, this is perhaps not the case because the test set is actually harder problem than the learning set, which probably is the reason for lower performance. Leaving aside the question if the performance shown in Fig. 2(d) ($r=0.54$) is indeed adequate to translate this into clinical practices (because the HRV itself is just a proxy for psychopathological conditions), the question of whether the model is overtraining needs to be addressed.

I suggest two things could be done to be more persuasive here, given the very low data setting:

(i) replace Fig. 2(c) with a leave-one-out cross validation results; i.e. take each data point out, train the model and test on the left out data. This will involve re-training the model as many times as you have training data. This will quantify the performance of your models on unseen data in the training domain and if, very low, confirm overtraining by the models. If not, it will persuade that the models are good and on the new task, the best we can hope for is the results shown.

(ii) given the test set is from a different / harder task, imagine a situation where we might be able to induce in the subject the harder task, but to a very limited amount; i.e. we cannot get enough training data in the harder task, but perhaps just enough to adapt the models learned on the easier two tasks to this new setting. This is the problem of transfer learning, where you learn in one setting (where you might have a lot of data) and transfer the learned model to a new setting in which there is a small amount of data, as done a lot in medical image processing problems (e.g. <https://arxiv.org/abs/1902.07208>) and with genomic data taken across different species (<https://www.biorxiv.org/content/10.1101/2019.12.26.888842v1>). For a task like this, this is relatively easy to do with a model like logistic regression.

ANSWERS TO THE REVIEWERS' COMMENTS

Reviewer #1

COMMENT/QUESTION: Rodrigues and colleagues tested the predictive value of newly developed virtual reality (VR) tools for subsequent heart rate variability (HRV) responses to acute stress. The authors report strong correlations between high-dimensional locomotor responses in specific VR environments with stress-related HRV.

This study represents an excellent example of how the translation of findings from the rodent literature (here, the open field and elevated plus maze tasks) to humans might work. The findings are novel and highly relevant for clinical contexts as they may pave the way to the development of non-invasive predictors for cardiovascular or mental disorders. A further strength of this study is the link between VR-based predictors and anxiety responses in a fear conditioning protocol tested on a separate day. I have no doubts that this study would make a unique contribution to the existing literature. I have only relatively minor comments.

Author's Reply: We thank the reviewer for his/her positive comments about our study and for the relevant proposals given below that have helped us improving our manuscript.

COMMENT/QUESTION: 1) I assume the authors collected also some baseline measurements of HRV. Did these baseline measures of HRV already predict HRV in the stressful situation (and potentially also pulse variability in the fear conditioning task)? If so, did the locomotor responses obtained in VR predict stress-related HRV beyond baseline HRV?

Author's Reply: Thank you for this question. A resting state baseline was not included in the study, as our original expectations had been that scenario 1 could have been used for this purpose; however, this turn out to be not the case as its novelty component activates SNS responses. Now, in order to see if we could address this question from the reviewer, we have calculated a 'baseline' measurement from the end of participants' exposure to the control relaxation condition towards the end of the study (BL period, Figure 4b). Since the baseline was recorded close to the stress task and intra-subject correlation at such close timing is expected, we performed correlational analyses with the other tasks; i.e., the dark maze and the fear conditioning task, the latter performed on a different day. This particular baseline correlated with both the HRV from the dark maze ($r = 0.45$, $p < 0.001$) but not with the PRV from the fear conditioning task ($r = 0.04$, $p = 0.712$). These correlations are clearly lower than those from our model's prediction and we include them here to reply to the reviewer's question. However, given that the baseline measurement does not correspond to a resting state, proper measurement, and on the sake of clarity, we prefer not to include these additional analyses in the manuscript.

COMMENT/QUESTION: 2) It is stated in the methods that only male participants were included. I assume this was motivated based on the rodent literature, which focusses also mainly on males. Nevertheless, the focus on males should be addressed as a potential limitation in the discussion.

Author's Reply: Indeed, as stated in the manuscript, this is our first study directly inspired by the rodent literature. The following note has been included in the Discussion as a potential limitation:

“Importantly, our study included only males given that our prediction linking behavioral and physiological responses to stress vulnerability was inspired in previous studies involving male rodents (Castro et al., 2012; Huzard et al., 2019) and, therefore, future work is warranted to assess the validity of our model to predict iHRV responses in females.”

COMMENT/QUESTION: 3) The authors focus exclusively on HRV, which is well justified. I was still wondering about the predictive value of the locomotor responses in the VR for other parameters that may be linked to psychopathologies. For instance, did the authors measure cortisol responses to the Montreal Imaging Stress test? If so, were these also predicted by the locomotor responses? More generally, would the authors expect that the obtained locomotor responses predict other arousal/stress measures beyond HRV?

Author’s Reply: We measured cortisol levels around the Stress task and addressed their potential prediction from a machine learning model in another lab project. However, in that case, we couldn’t fit and validate a model based on locomotor responses that predicts corticosterone stress responsiveness. It is true that it was only a side project and that we may not have dedicated enough resources to fully exclude that behavioral/locomotor responses from novelty tests can predict stress-triggered cortisol responses. Future studies would be needed to address this question fully and specifically.

Regarding the second question, we have now calculated whether our model can predict stress-induced cortisol responses and found just a low and non-significant correlation between our model’s prediction and the cortisol response (Spearman’s $\rho = 0.16$, $p=0.098$). Due to the inefficacy of modeling the cortisol response with locomotor activity, we decided not to include cortisol data to have a greater focus on HRV.

Regarding the predictive capability of locomotor responses for other arousal/stress measures, in this study we decided to focus mainly in predicting HRV and HR. Since the RMSSD formula was the most predominant in the iHRV index, which was used to train our model, and also the one that correlated higher with the model’s predictions, we consider this model to be capturing the parasympathetic influence. Indeed, we think other physiological markers of arousal / stress could be studied, for example, the sympathetic response measured from electrodermal activity features (like the number of skin conductance responses, skin conductance level change). Future studies should also address this question.

COMMENT/QUESTION: 4) I was surprised that the locomotor responses were not at all correlated with trait or state anxiety measures. How do the authors explain the absence of such correlations? What are the implications of the lack of such correlations for the conclusions of the authors and for the potential use of the locomotor responses as a predictive tool for later psychopathology?

Author’s Reply: Indeed, we anxiety measures are not correlated with the model (i.e., composed of variables based on locomotor responses) that predicts specifically iHRV responses. In fact, another ongoing study in our lab is aiming at developing a model predictive of trait anxiety based on locomotor responses to three challenging scenarios. What this new study is revealing is that trait anxiety is predicted by a different combination of locomotor parameters.

Therefore, our findings (present study and our ongoing one) indicate that there is a high level of precision in what the developed machine learning models capture regarding specific physiological reactivity or behavioral predispositions (note that this is in line with the important variation existing in people's behavioral and physiological responses to stressful challenges), supporting the view that our current model reported here optimizes prediction for stress-induced iHRV/cardiovascular responses. Importantly, anxiety (trait or state) measurements with the STAI inventory typically explain a much lower percentage of the variance in stress reactivity than the high explanatory capacity achieved by our machine learning model in this study.

Regarding the implications for the potential use of the iHRV model developed here regarding its use as a predictive tool for later psychopathology, our high predictive capacity is supportive of potentially better prediction for later psychopathology than existing tools/questionnaires. In our view, it will be important to benchmark it in longitudinal studies. Likewise, we are planning to investigate its capacity to predict psychopathology in clinical populations.

To address these referee's comments, we have added the following sentences to the Discussion section:

"Self-reported anxiety measurements, as obtained with the STAI inventory, typically explain a lower percentage of the variance in stress reactivity (see, e.g., Goette et al., 2015) than the one achieved by our machine learning model. Therefore, the lack of correlation between anxiety measures and our model predictions supports the view that our model has achieved a high level of precision in capturing specific physiological reactivity (as opposed to other non-related behavioral predispositions), further supporting its robustness in predicting stress-induced iHRV/cardiovascular responses."

The following part was added later in the Discussion section:

"In addition, although our results indicate a potential high predictive capacity of our model for later psychopathology than existing tools/questionnaires, it will be important to benchmark it in both longitudinal studies and clinical populations."

COMMENT/QUESTION: 5) In the methods, it was not entirely clear to me how participants were assigned to the training or test data sets. Was this solely based on the mentioned anxiety related criteria? May the definition of the training vs. test samples based on anxiety-related criteria have affected the potential predictive value for state and trait anxiety?

Author's Reply: The training set contained participants who presented low (STAI-T<35) and moderately high (STAI-T>45) scores to ensure that we had a wide range of HRV values (since trait-anxiety might influence HRV), and that the model wouldn't be biased and towards a specific trait-anxiety range of participants. The test set contained participants with moderate (35 > STAI-T > 45) scores. This information was presented on the Results section and has now been edited for improved clarity, as follows:

"Given that trait anxiety (as measured with the Trait questionnaire from the State-Trait Anxiety Inventory (STAI-T) (Spielberger, 1983)) can be associated with HRV⁵⁴, the training/discovery set contained participants who presented low (STAI-T<35) and moderately high (STAI-T>45) scores to

ensure that we had a good representation of participants' iHRV in each of the two datasets, and to prevent biasing HRV prediction on trait anxiety."

This information is now also present in the Methods section:

"The training/discovery set contained participants who presented low (STAI-T<35) and moderately high (STAI-T>45) scores. The test/replication set contained participants with moderate (35 > STAI-T > 45) scores."

Regardless of the partition (all subjects, only low STAI-T, only intermediate STAI-T, only moderate-high STAI-T), note that we never observed a significant association between trait- or state-anxiety and HRV (RMSSD) in the dark maze or stress test.

COMMENT/QUESTION: 6) Related to the previous comment: was the assignment of participants to the training vs test data sets varied and did the authors use some form of permutation test to test the robustness of the observed effects? Alternatively, one might also think about a variant of the 'leave-one-out' methodology?

Author's Reply: We did not try other partitions of train and test sets, since we had a fixed criterion for the partitions being used to ensure spread of participants background in the different training and testing batches (see text). Following the reviewer's suggestion, we have now performed a 10-fold cross-validation in the training dataset as recommended in the Scikit-learn manual for cross validation (<https://scikit-learn.org/stable/modules/crossvalidation.html>: "*As a general rule, most authors, and empirical evidence, suggest that 5- or 10- fold cross validation should be preferred to leave one out*"; even though this recommendation might not be as important for our sample size). We have added this to the Supplementary section "Model validation on the training dataset" and Figure S3b.

COMMENT/QUESTION: 7) Finally, the authors suggest that their VR protocol might be used as a predictive tool for cardiovascular diseases or mental disorders and I completely agree that this protocol may be helpful in this respect. Perhaps the authors could still elaborate in the discussion on the (dis)advantages (pros and cons) of their protocol relative to simply using another standardized stress protocol such as the TSST (PMID: 8255414) or the SECPT (PMID: 18403130).

Author's Reply: We thank the reviewer for this suggestion. To address his/her comment, we have now introduced the following sentences in the Discussion section:

"On the other hand, as compared to non-VR based laboratory stress challenges, such as the trier stress test (TSST; Kirschbaum et al., 1993) or the socially evaluated cold pressor test (SECPT; Schwabe et al., 2008), our IVEs constitute a more standardized procedure allowing for more reproducible and controlled conditions across subjects and laboratories. In the future, it will be important to benchmark the capacity of our IVEs to both, trigger physiological responses and predict stress vulnerability against procedures such as the TSST or SECPT."

Reviewer #2

COMMENT/QUESTION: Parsing heterogeneity in stress responsivity is becoming an increasingly important research goal. This paper therefore represents an important contribution as it presents interesting data towards accurate characterization of resilience versus sensitivity to stress with implications for the implementation of neuroscientific findings into clinical care. In order to be suitable for publication in Nature Communications, however, some issues need to be clarified. The comments below are intended to assist the authors in sharpening a future revision of their paper.

Author's Reply: We thank the reviewer for his/her positive comments about our study and for the relevant proposals given below that have helped us improving our manuscript.

COMMENT/QUESTION: -In the Introduction, the sentence: "Stress-related cardiovascular reactivity values elicited in the laboratory setting have been shown to outperform the predictive capacity of resting assessments in studying disease development and have been found to be stable personal characteristics that are fairly consistent across time and between stressors" is not completely correct. In the field of cardiovascular disease, the so-called reactivity hypothesis has failed to fulfill its promise leading to inconsistent findings (see for example the Whitehall II Study; Carroll et al., 2001). The authors should acknowledge that cardiovascular responses to the brief and unecological stressors that are commonly used in the lab have serious limits that have been overcome by more expensive and time-consuming ecological ambulatory studies (e.g., studies by Pickering, Gerin, Shapiro, etc.). This is in my opinion, a major advantage of the proposed study which makes an attempt to take ecological stressful situations into the lab. In fact, the authors do not use cardiovascular reactivity as a predictor, but a complex combination of behavioral features (i.e. 172 behavioral features to start with). Please revise the Introduction to make this point clearer.

Author's Reply: We thank the reviewer for these important remarks. We have now implemented the commentary in the Introduction to highlight all those aspects, as follows:

"However, in the long-term, the prognostic value of stress reactivity studies has proven to be quite modest (Gerin et al., 2000; Carroll et al., 2001; Kamarck and Lovallo, 2003). typically based on exposure to short and non-ecologically-relevant laboratory stressors, and relying on a handful of variables. Although these limitations can be overcome by exposure to more complex challenges and performing ambulatory cardiovascular measurements (Ottaviani et al., 2006, 2011; Bailey et al., 2019), these approaches are rather expensive, time-consuming and not easy to standardize for inter-individual comparisons."

COMMENT/QUESTION: - "To define stress vulnerability, we focused on heart rate variability (HRV) a surrogate index of cardiac vagal break that yields information about autonomic flexibility". Could the authors please better introduce the reasons why they have chosen this particular index as a measure of stress vulnerability? There are good reasons for this choice, but in my opinion this rationale should be better explained.

Author's Reply: We have now tried to provide further information about the reasons to focus on HRV, and the following sentences added to the Introduction (in addition, please, note that we give a longer introduction in Supplementary Information to the mechanisms underlying HRV and its implications):

“In addition, resting state HRV reflects the degree of flexible control exerted by the prefrontal cortex over the periphery (Ottaviani, 2018). In recent assessments of the literature, high HRV levels during resting conditions has been identified as a top biomarker for stress resilience (Walker et al., 2017; Carnevali et al., 2018). Conversely, [...] See Supplementary Introduction for further details on the mechanisms underlying HRV and the effectiveness of HRV as a biomarker in stress-related psychopathologies.

COMMENT/QUESTION: - In the Introduction “Low HRV values are associated with increased risk of ...” the authors should distinguish between tonic and phasic HRV, which provide different information and have different prognostic value. This distinction should be made clearer throughout the manuscript.

Author’s Reply: We thank the reviewer for this suggestion. Sometimes, the statements include many studies carried out in different experimental conditions. Thus, we have tried to provide this information throughout the text in the most precise manner possible. In both Introduction and Discussion, at the appropriate levels, we have added “tonic” or “phasic” as appropriate.

COMMENT/QUESTION: -When they introduce the behavioral features (“The most discriminating features included positioning and gait parameters”), the authors should briefly describe the meaning of such features to help the reader’s understanding. These are mentioned in the Discussion (“e.g., minimum distance to the corners of the empty room, ratio between time in the center and time in the periphery of the empty room, and time spent on the narrowest ledge of the elevated alley”) but in my opinion it would be useful to move this section earlier in the text.

Author’s Reply: In the Results section, this is reflected in the following sentence, with the corresponding explanations, as follows:

*“The most discriminating features included positioning and gait parameters, with the largest contribution from the former category, including minimum distance to the corners of the empty room [*Corner dist (er)*], vertical acceleration in the elevated alley [*Vert acc (ea)*], and ratio between time in the center and time in the periphery of the empty room.”*

COMMENT/QUESTION: - I may have missed it, but is there an association between the identified behavioral features and self-reported trait/state anxiety?

Author’s Reply: Indeed, this is an interesting question on its own and, indeed, we have been studying possible associations between locomotor parameters and self-reported trait- and state-anxiety in another ongoing study in our lab. That study is aiming at developing a model predictive of trait anxiety based on locomotor responses to three challenging scenarios. What this new study is revealing is that trait anxiety is predicted by a different combination of locomotor parameters (i.e., not precisely the same parameters that define the current iHRV model), and will be the focus of our next manuscript, currently in preparation.

COMMENT/QUESTION: -The sentence “Although anxiety disorders are frequently accompanied by alterations in HRV, in our nonclinical population, none of the trait or state anxiety or threat anxiety

measures predicted iHRV. This finding further supports the robustness of our model in predicting cardiovascular stress vulnerability” is not clear to me. Why this further supports the robustness of the model?

Author’s Reply: Thank you for this remark. We agree that robustness is misleading since we want to stress that our model is specific to cardiovascular responses (and not personality traits / states). Our findings indicate that there is a high level of specificity in what the developed machine learning model capture regarding specific physiological reactivity. In line with the important variation existing in people’s behavioral and physiological responses to stressful challenges), supporting the view that our current model reported here optimizes prediction for stress-induced iHRV/cardiovascular responses. Importantly, anxiety (trait or state) measurements with the STAI inventory typically explain a much lower percentage of the variance in stress reactivity than the high explanatory capacity achieved by our machine learning model in this study.

To address these referee’s comments, we have added the following sentences to the Discussion section, and in addition we have replaced *robustness* with *specificity*:

“Self-reported anxiety measurements, as obtained with the STAI inventory, typically explain a lower percentage of the variance in stress reactivity (see, e.g., Goette et al., 2015) than the one achieved by our machine learning model. Therefore, the lack of correlation between anxiety measures and our model predictions supports the view that our model has achieved a high level of precision in capturing specific physiological reactivity (as opposed to other non-related behavioral predispositions), further supporting its specificity in predicting stress-induced iHRV/cardiovascular responses.”

COMMENT/QUESTION: -Participants section (Method): were participants asked to refrain from caffeine, alcohol, strenuous exercise, etc. before the experimental session?

Reply: Thank you for this remark. Indeed, participants were asked to refrain from eating or drinking (except water) one hour before the experiment took place. This is now added to the methods section.

COMMENT/QUESTION: -Participants are all males. Please comment on this choice which prevents generalization of results to the general population.

Author’s Reply: Indeed, as stated in the manuscript, this is our first study directly inspired by the rodent literature. The following note has been included in the Discussion as a potential limitation:

“Importantly, our study included only males given that our prediction linking behavioral and physiological responses to stress vulnerability was inspired in previous studies involving male rodents (Castro et al., 2012; Huzard et al., 2019) and, therefore, future work is warranted to assess the validity of our model to predict iHRV responses in females.”

COMMENT/QUESTION - Minor:

Page 3, line 110: “responses progressively habituated during immersion VR in each scenario”. I think that there is a typo and that “VR” should be deleted.

Author’s Reply: Thanks for noting this. The typo has been corrected.

Reviewer #3

COMMENT/QUESTION: In this paper, the authors have developed three VR scenarios with locomotion tasks for phenotyping inspired by the animal literature: (i) open room, (ii) elevated alley, and (iii) dark maze corridor and computed individuals’ HRV responses to stress. With the help of machine learning methods, they show the predictive power of high-dimensional locomotor responses to predict HRV responsiveness during stress exposure. Their model’s predictions generalized to a different cohort and to other stressful challenges and outperform other stress prediction instruments, such as anxiety questionnaires. To my knowledge, no study has demonstrated the prognostic value of behavioral data as a predictor of stress vulnerability using objective (i.e., not self-reported) stress measurements. Hence, the work provides a significant research contribution to the field of locomotion research and prediction of cardiovascular responsiveness to subsequent stressful situations.

The paper addresses an important topic, but there are also several issues. Overall, I think the findings are not very surprising. In previous work stress-related cardiovascular reactivity values elicited in the laboratory setting have been shown to outperform the predictive capacity of resting assessments in studying disease development. The authors aimed to replicate this by developing immersive virtual environments (IVEs) to predict physiological stress susceptibility. While the ML algorithms perform greatly to the training data (as expected), the correlation for the test/replication dataset is rather low. The authors discuss in more details the reasons for the rather low correlation and discuss the findings from the previous work on stress-related cardiovascular reactivity values elicited in the laboratory setting.

Author’s Reply: We thank the reviewer for his/her positive comments about our study and for the relevant proposals given below that have helped us improving our manuscript.

In reply to the comments above, we would like to note that our correlations are, in fact, quite strong in the context of both the individual differences literature and current machine learning developments.

COMMENT/QUESTION: Next, the authors argue that it is important to develop tools that identify predictive markers of individual differences in HRV responsiveness without exposing subjects to high stress. While I can follow this argument in principle, it would be good if authors could discuss the motivation in a bit more detail.

Author’s Reply: We thank the reviewer for noting that the motivation to develop a diagnostic tool that predicts vulnerability to a highly stressful challenge without the need to exposing individuals to high stress was not well motivated, we have now added the following motivation to the argument/sentence:

“Importantly, given that experimental exposure to highly stressful conditions may be detrimental to health in susceptible individuals, ...”

COMMENT/QUESTION: Furthermore, there are several missing details about the experimental setup and procedure, which need to be clarified in the revision. For instance, which HMD was used? HTC Vive Pro / Eye? Which computer was used for rendering? How was the tracking implemented? Which VR system (Unity3D?) was used to render the VE? How much end-to-end latency is induced by the system. The authors describe that subjects could never see the physical room to increase immersion and the sense of novelty. I think the authors mean a sense of presence. Also, the machine learning algorithm is rarely explained. Was this based on a CNN?

Author's Reply: Thank you for this suggestion. We have now added the following information regarding hardware and rendering to the Methods section:

“Virtual reality was performed using a commercial virtual reality system developed by HTC and Valve (HTC Vive). The head mounted display (HMD) presents stereoscopic images of the virtual scene to the participant with a resolution of 1080x1200 pixels per eye and a refresh rate of 90 Hertz. The tracking relies on two lighthouse stations sweeping structured light lasers into the testing room. The HMD uses several sensors (laser position sensors, microelectromechanical sensors, gyroscope and accelerometer) to reliably infer their position and orientation in 3D space in real-time and with sub-millimeter precision. The different 3D scenes were rendered in Unity3D (www.unity3d.com) running on a computer dedicated for VR and motion capture, equipped with a Core i7 CPU clocked at 4.0GHz, 16GB of main memory and a GeForce GTX TITAN X graphics card. 3D-scenes were modelled in Blender (<https://www.blender.org>) and imported into Unity3D. The experimental logic was programmed in C# within Unity3D. The app-to-display latency for the HTC Vive running Unity3D apps in a computer with a similar configuration to the one used in this experiment has been determined to be on average 31.33 ms with a standard deviation of 1.41 ms (Chénéchal and Chatel-Goldman, 2018).”

Indeed, presence is the correct term since it reflects the “sense of being in the virtual environment” and while immersion entails technical elements associated with realism. Consequently, we have replaced this term, thank you for noticing it. We cannot however make a strong claim that presence is actually increased since we have never tested the effect of seeing or not the room before. Anecdotal reports have led us to start running experiments in this manner and we do not think it hinders the experimental quality.

Regarding the machine learning algorithm, it is not a CNN. It is based on boosting decision trees. It is an ensemble method that uses several decision trees to make a prediction. This method is well known for obtaining winning solutions in various data competitions and has been increasingly used in the medical field, amongst other domains. We have now explained this method with more detail in the methods section.

“XGBoost⁵⁵ is a new implementation of the gradient tree boosting technique that has been tested in different datasets; it achieves high accuracy and requires much less computation time than deep neural nets⁸⁹. It is known for obtaining winning solutions in various data competitions. Its authors reported that “among the 29 challenge-winning solutions published on Kaggle’s blog during 2015, 17 winning solutions used XGBoost.”⁵⁵. XGBoost has also been applied to the medical field (Gao et al., 2018; Nishio et al., 2018; Qiao et al., 2018). Here we used XGBoost to predict iHRV during the persistent threat scenario with the set of selected behavioral features. We used the XGBRegressor function from the Python XGBoost package to fit our model. There are several adjustable hyperparameters in XGBoost. In this study, the step size shrinkage (η), maximum

depth of tree (max.depth), minimum sum of instance weight (min.child.weight), and maximum number of iterations (nrounds) were optimized with Bayesian optimization (Shahriari et al, 2015) using the Python package hyperopt. The performance of our model's predictions was evaluated by correlating the predicted with the true value of the parasympathetic PC on a holdout sample (not used in model learning)."

COMMENT/QUESTION: Finally, I think the authors should explain in more detail why they decided on the provided scenarios, which only partly mimic the setups from the animal literature, and which could have been chosen very differently. Were the scenarios randomized between subjects? If not, why?

Author's Reply: Similar to what is customary in our rodent laboratory during individual phenotyping, exposure to the two scenarios was not randomized between subjects. Our goal is to have a testing pipeline that allows us to obtain meaningful measurements to characterize our individuals, not to compare the diagnostic value of each of the scenarios separately. In order to address the reasons why we adapted the rodent tests to our VR versions, we have now added the following information to the text / Results section:

"Although both mimicked key aspects of tests from the rodent literature (see below), our goal was to adapt the scenarios to standard laboratory room dimensions (i.e., 3.5 m x 6 m) which determined that large spaces or redundant aspects of the rodent tests were simplified to their gist."

COMMENT/QUESTION: As a minor remark there is some inconsistency throughout the paper, e.g., "head-mounted display" vs. "head mounted display", they introduced the acronym HMD, but never used it etc.

Author's Reply: We thank the reviewer for noting the inconsistency in this terminology, which we have now corrected throughout the text.

COMMENT/QUESTION: To summarise, I think the paper lacks too many details and discussions of the results, which need to be added before the paper can be accepted. However, the findings appear to be a significant research contribution and very interesting, hence, I would recommend a major revision.

Author's Reply: We thank the reviewer again for all the comments and suggestions and hope that our revision has now improved the manuscript in all those aspects that were not sufficiently clear in the previous version.

Reviewer #4

COMMENT/QUESTION: This paper is about predicting physiological vulnerability in response to stress. It builds on observations from animal studies where it is shown that behaviors inducing anxiety and reactions to novelty can subsequently explain the induction of susceptibility to depressive behaviors.

The novelty in the work presented here is to build on it for human subjects, but carefully design virtual reality environments via which stressful stimuli may be induced, the response to such stress quantified in a high dimensional space by features extracted from wearable sensor measurements and subsequently predict behavior in a previously unseen but a more severe task. The work is imaginative in setting up the VR environments, using heart rate variability as proxy for induced physiological state and careful in decoupling any systematic confounding effects (e.g. respiration). The manuscript is clearly written and easy to follow. Figure captions are sufficiently detailed and informative. As such, I feel positive towards recommending the work for publication.

Author's Reply: We thank the reviewer for his/her positive comments about our study and for the relevant proposals given below that have helped us improving our manuscript.

COMMENT/QUESTION: However, I have a major reservation that needs to be addressed before publication, which has to do with the results obtained via machine learning, clearly apparent in Fig. 2(c) and (d) as a huge disparity between predictions made on the training set and the test set. This will usually be seen as an indication of overtraining in usual machine learning problems, where the training and test sets arise from the same joint distribution (of inputs and responses). In the problem considered here, however, this is perhaps not the case because the test set is actually harder problem than the learning set, which probably is the reason for lower performance. Leaving aside the question if the performance shown in Fig. 2(d) ($r=0.54$) is indeed adequate to translate this into clinical practices (because the HRV itself is just a proxy for psychopathological conditions), the question of whether the model is overtraining needs to be addressed.

I suggest two things could be done to be more persuasive here, given the very low data setting:

(i) replace Fig. 2(c) with a leave-one-out cross validation results; i.e. take each data point out, train the model and test on the left-out data. This will involve re-training the model as many times as you have training data. This will quantify the performance of your models on unseen data in the training domain and if, very low, confirm overtraining by the models. If not, it will persuade that the models are good and on the new task, the best we can hope for is the results shown.

(ii) given the test set is from a different / harder task, imagine a situation where we might be able to induce in the subject the harder task, but to a very limited amount; i.e. we cannot get enough training data in the harder task, but perhaps just enough to adapt the models learned on the easier two tasks to this new setting. This is the problem of transfer learning, where you learn in one setting (where you might have a lot of data) and transfer the learned model to a new setting in which there is a small amount of data, as done a lot in medical image processing problems (e.g. <https://arxiv.org/abs/1902.07208>) and with genomic data taken across different species (<https://www.biorxiv.org/content/10.1101/2019.12.26.888842v1>). For a task like this, this is relatively easy to do with a model like logistic regression.

Author's Reply: Thank you for your relevant suggestions

(i) We understand the reviewers' concern; however, as a reminder, we used different samples, with different characteristics, for training (subjects scoring either high or low in trait anxiety scores) and for testing (subjects with intermediate scores of trait anxiety). This approach excludes that our results on

the testing group reflect any overtraining. Despite this, and to further address the Reviewer's proposal, we have now performed a 10-fold cross validation scheme as recommended in the Scikit-learn manual for cross validation (https://scikit-learn.org/stable/modules/cross_validation.html): "As a general rule, most authors, and empirical evidence, suggest that 5- or 10- fold cross validation should be preferred to leave one out"; even though this recommendation might not be as important for our considerable sample size). The outcome of this analysis is now included in Figure S3 and supplementary section "Model validation on the training dataset". We also computed other evaluation metrics to compare the predictions between the test set and the cross-validated training set and added them to the Supplementary Information section, in the section related to "Model validation on the training dataset".

Therefore, these analyses indicate that the training set represent a more difficult task than the testing set, which is explained by the characteristics of the sample (i.e., two extremes of trait anxiety in the training set) representing separate sets of data, at difference to the more continuous character of the testing dataset (i.e., subjects with intermediate scores of trait anxiety). Accordingly, training the model in the extremes of trait anxiety (less prevalent in the general population) offered enough variability of behavior and autonomic responses for a successful prediction in the intermediate trait anxiety sample (more prevalent in the general population). Furthermore, we also assessed whether the hyperparameters of the model could be optimized at each cross-validation run in the training set, and if this would increase the cross-validation performance, which lead to a correlation of 0.54.

(ii) As indicated above, in fact the test set is not a harder task, and therefore the proposed solution would not be applicable in this case. However, the referred articles raise indeed an exciting approach for future studies relevant to the approach undertaken here, in which we adapt tests from the animal literature to probe their relevance in human populations. Sparked by this interesting suggestion, we have now added the following sentences to the Discussion section:

"Additionally, our model allows training continuation by iteratively training with new observations which allows us to encompass a larger representation of different traits. Furthermore, approaches based on transfer learning can also enable the adaptation of our model to other target domains such as clinical settings or even, in a similar fashion to mapping biological relationships from mouse to human (Stumpf et al, 2019), to facilitate translational approaches mapping animal behavioral/physiological interplay to human models or vice-versa."

Reviewer #1 (Remarks to the Author):

The authors responded appropriately to all of my previous comments and the revision has further strengthened the manuscript. I have no further comments.

Reviewer #2 (Remarks to the Author):

It is my impression that the manuscript has been strengthened in a number of regards. The authors responded conscientiously to each comment and were also very responsive to the criticism raised by the other reviewers. In my opinion, the manuscript does now meet the high standards for publication in Nature Communications.

Thank you for giving me the opportunity to be a reviewer for this manuscript.

Reviewer #3 (Remarks to the Author):

The authors have carefully addressed my concerns and gave detailed feedback regarding the review's comments.

In particular, they have tried to calculate a 'baseline' measurement from the end of participants' exposure to the control relaxation condition towards the end of the study. They also performed correlational analyses with the remaining tasks. However, since the baseline measurement does not correspond to a resting state, I agree that this should better not be included in the paper.

Unfortunately, the presented model could not predict stress-induced cortisol responses. However, I acknowledge that the authors performed the correlation test to check.

Furthermore, I acknowledge that the authors have added sufficient detail regarding hardware and rendering to the Methods section. Moreover, the explanation of the applied ML methods is much clearer now, and I think that it is good shape.

The clarification of the confusing usage of the terms "presence" and "immersion" is also acceptable now.

I also read the other reviewers' comments and the answers of the authors.

I think that most concerns and criticisms have been addressed in an appropriate way. Points, which have not been addressed in the manuscript, have been explained and authors have argued why they prefer to not include them in the manuscript. I really acknowledge the extra work that the authors have performed with the additional data analyses.

One open aspect appears to be the concerns by reviewer #4 regarding the overtraining of the machine learning and the transferred learning issue. Since this is not my field of expertise, reviewer #4 would have to evaluate if the authors' response is acceptable.

To summarize, I am in favor of accepting the manuscript in its given form and thank the authors for their interesting work.

Reviewer #4 (Remarks to the Author):

In the revised manuscript the authors have included cross validation result on the training set they used (Supplementary Fig. S3 and text leading to it) with a correlation in the region of 0.54 between predictions and measurements. This does show that the study is not dealing with an over-trained system for its predictions, adequately addressing the concern I raised in the review. I think the manuscript could now be published.